# Applying Flow Virometry to Study the HIV Envelope Glycoprotein and Differences Across HIV Model Systems

**DOI:** 10.3390/v16060935

**Published:** 2024-06-09

**Authors:** Jonathan Burnie, Claire Fernandes, Ayushi Patel, Arvin Tejnarine Persaud, Deepa Chaphekar, Danlan Wei, Timothy Kit Hin Lee, Vera A. Tang, Claudia Cicala, James Arthos, Christina Guzzo

**Affiliations:** 1Department of Biological Sciences, University of Toronto Scarborough, Toronto, ON M1C 1A4, Canada; jonathan.burnie@mail.utoronto.ca (J.B.); claire.fernandes@mail.utoronto.ca (C.F.); ayuship0606@gmail.com (A.P.); arvin.persaud@mail.utoronto.ca (A.T.P.); deepa.chaphekar@mail.utoronto.ca (D.C.); leetim.lee@mail.utoronto.ca (T.K.H.L.); 2Department of Cell and Systems Biology, University of Toronto, Toronto, ON M5S 3G5, Canada; 3Department of Immunology and Infectious Diseases, Harvard T.H. Chan School of Public Health, Boston, MA 02115, USA; 4Laboratory of Immunoregulation, National Institute of Allergy and Infectious Diseases, National Institutes of Health, Bethesda, MD 20892, USA; weida@niaid.nih.gov (D.W.); ccicala@niaid.nih.gov (C.C.); jarthos@niaid.nih.gov (J.A.); 5Flow Cytometry and Virometry Core Facility, Department of Biochemistry, Microbiology, and Immunology, Faculty of Medicine, University of Ottawa, Ottawa, ON K1H 8M5, Canada; vtang@uottawa.ca; 6Department of Immunology, University of Toronto, 1 King’s College Circle, Toronto, ON M5S 1A8, Canada

**Keywords:** human immunodeficiency virus (HIV), calibrated flow virometry, molecules of equivalent soluble fluorophore (MESF), nanoscale flow cytometry, gp120/gp41, HIV Env, HIV trimer, Env conformation, virion capture, neutralization

## Abstract

The HIV envelope glycoprotein (Env) is a trimeric protein that facilitates viral binding and fusion with target cells. As the sole viral protein on the HIV surface, Env is important both for immune responses to HIV and in vaccine designs. Targeting Env in clinical applications is challenging due to its heavy glycosylation, high genetic variability, conformational camouflage, and its low abundance on virions. Thus, there is a critical need to better understand this protein. Flow virometry (FV) is a useful methodology for phenotyping the virion surface in a high-throughput, single virion manner. To demonstrate the utility of FV to characterize Env, we stained HIV virions with a panel of 85 monoclonal antibodies targeting different regions of Env. A broad range of antibodies yielded robust staining of Env, with V3 antibodies showing the highest quantitative staining. A subset of antibodies tested in parallel on viruses produced in CD4^+^ T cell lines, HEK293T cells, and primary cells showed that the cellular model of virus production can impact Env detection. Finally, in addition to being able to highlight Env heterogeneity on virions, we show FV can sensitively detect differences in Env conformation when soluble CD4 is added to virions before staining.

## 1. Introduction

Despite decades of research dedicated to studying the human immunodeficiency virus (HIV), an efficacious preventative vaccine remains elusive. One of the prime reasons for this is due to the difficulty that exists with targeting the viral envelope glycoprotein (Env) [1,2,3,4]. Env has been highly studied due to its critical roles in mediating HIV entry through binding and fusion with CD4^+^ cells (reviewed in [5,6,7]). Early work in this field identified Env as a trimer, consisting of heterodimers of the surface unit, gp120, and the transmembrane subunit, gp41 [5,7]. As the sole viral protein on the HIV surface, Env is a key antigen for anti-HIV immune responses and for the development of an effective HIV cure. Unfortunately, targeting Env has proved challenging due to its heavy glycosylation, high genetic variability, conformational flexibility, and its low abundance on virions [1,8,9,10,11,12,13,14]. Despite this, Env continues to hold promise as a viable target for vaccine strategies, and thus a need to better understand this protein remains. 

The variability of Env is particularly well documented, with extensive variation among different viral strains and five hypervariable regions of gp120 interspersed among more constant regions of the protein [15,16,17]. While individuals mount antiviral responses that produce antibodies targeting the trimer [18,19], these antibodies alone fail at controlling infection [3] because of the trimer’s glycan shielding and high propensity to accumulate mutations. However, a subset of individuals generate broadly neutral antibodies (bNAbs) with notable potency and breadth [20,21,22,23]. These antibodies have helped bring forth a better understanding of Env structure and have been characterized to target distinct epitopes on the trimer, including the following: the CD4 binding site (CD4bs), CD4 inducible epitopes (CD4i), glycan-dependent regions at the first and second variable (V1V2) loop apex and third variable loop (V3), gp41, and the membrane proximal region (MPER) [17,24,25,26]. While the promise of bNAbs in managing infection in vivo is an area of ongoing investigation [25,26,27], these antibodies have proven to be a powerful tool in the laboratory, where they can be used in antibody-based assays such as neutralization and immunoprecipitation assays. Although such antibody-based techniques have been routinely employed in the HIV field with high success, the use of antibody techniques that assess viruses at a single particle level is less common.

Flow virometry (FV), or flow cytometry applied to viruses, is an emerging technique that can be used to help bridge the gap in this field. Flow virometry was originally coined in 2013 when Arakelyan et al. studied the surface proteins of HIV with flow cytometry using viruses bound to magnetic nanoparticles [28]. Since then, their group has continued to develop this methodology for use on HIV and Dengue virus [29,30,31,32]. While flow cytometry has been utilized for decades to enumerate viruses in the marine biology field [33,34,35], visualization of viruses on conventional cytometers often requires labelling with a fluorescent protein or tag [36,37,38,39,40], since virions typically fall within the range of instrument background noise when using light scatter for detection based on their size. More recently, advances in flow cytometry instrumentation have allowed for viruses and vesicles in the 100 nm range to readily be detected on high-sensitivity cytometers [41,42,43,44,45,46]. With these advances, our lab and others have moved towards staining viral surface antigens on particles directly in cell culture supernatants, without the requirement for magnetic beads or fluorescently tagged labels to detect virions [41,42,43,44,47,48,49,50,51,52,53].

Since FV can provide sensitive, quantitative measurements of surface proteins on single virions [47,54], we sought to use this technique to study HIV Env produced from different cell types. While FV has been used to study HIV Env in select studies [55,56], a protocol that stains native virions directly in cell culture supernatants, without the requirement for wash steps and ultracentrifugation, has not been performed, and would advance the utility of this technique in a significant way. Herein we demonstrate these advances with flow virometry, and provide sensitive assessments of HIV Env with a variety of anti-Env monoclonal antibodies (mAbs), including the broadly neutralizing mAbs PG9 and PGT126, and non-neutralizing mAbs such as 246-D. Further, by comparing how anti-Env antibodies perform in FV versus virus neutralization and virus capture assays, we showed that distinct differences exist in how the same antibodies perform across these assays. Additionally, we show that the same virus isolate (HIV_BaL_) produced in different cell types can yield differential Env labelling, indicating that the virus producer cell can impact the Env quality, quantity, and/or accessibility on the virus particle. Finally, by staining viruses in the presence of soluble CD4, we show that FV can sensitively assess differences in the HIV envelope trimer conformation, demonstrating an added utility of FV for studying HIV Env.

## 2. Materials and Methods

### 2.1. Cell Culture

The T cell lines (PM1, H9, Jurkat E6-1) and peripheral blood mononuclear cells (PBMC) used to produce virus through infection with replication-competent virus isolates were maintained in RPMI-1640 (Wisent, Saint-Jean-Baptiste, QC, Canada, Cat#350-000-CL) with 10% heat-inactivated fetal bovine serum (Wisent, Cat#098150), 100 U/mL penicillin, and 100 µg/mL streptomycin (Life Technologies, Carlsbad, CA, USA, Cat#15140122). All cells were grown in a 5% CO_2_ humidified incubator at 37 °C. Primary cells were collected through the NIH Department of Transfusion Medicine protocol that was approved by the Institutional Review Board of the National Institute of Allergy and Infectious Diseases, National Institutes of Health. Additional primary cells were isolated from whole human blood collected from self-declared healthy volunteers in agreement with the University of Toronto Research Ethics Board approval (Human Protocol #00037384). Informed consent was written and was provided to all study participants.

### 2.2. Virus Production

For infection of T cell lines, cells were pelleted and resuspended in 1 mL of replication-competent virus isolates (BaL or IIIB) for 4 h. Afterwards, fresh media was added to the cells until the time of harvest. For infection of primary cells, PBMC were activated with anti-CD3, IL-2 (20 IU/mL), and retinoic acid (10 nM) for 3 days before infection. Activated PBMC cultured in 6 well plates were infected with 5 ng of p24 of virus stocks. During infection, every 3 days, fresh media was used to replace half of the media in the wells. Cell culture supernatants containing virus were harvested 7–12 days later based on viral titre (p24 levels). HIV pseudoviruses and infectious molecular clones (IMC) were produced through transfection of HEK293T cells using Polyjet In Vitro Transfection Reagent (FroggaBio, Toronto, ON, Canada, Cat#SL100688), as described previously [48]. Pseudoviruses were generated using 2 μg of SG3ΔEnv plasmid (ARP-11051) as the viral backbone, and 1 μg of HIV-1 BaL.01 Env plasmid (ARP-11445). Infectious viruses generated through transfection with IMCs was performed with 3 μg of NL4-3 (ARP-114) or NL4-3 BaL (ARP-11441) pDNA, respectively. All virus-containing culture supernatants were centrifuged for 5 min at 300× *g* to remove cellular debris before being aliquoted and stored as viral stocks at −80 °C until use in subsequent assays, without any filtration of the supernatants.

### 2.3. Flow Virometry

Flow virometry was performed using a Beckman Coulter CytoFLEX S with a standard optical configuration. The PE gain and threshold optimization for detection of virus and calibration beads were performed as described previously [48]. All samples were acquired for 30 s at a sample flow rate of 10 μL/min, except for PBMC viruses, which were acquired for 2 min to allow for enhanced visualization of the virus populations. For all labelling, crude, cell-free supernatants containing virus were stained undiluted, with an average particle concentration of ~10^8^ particles/mL. All staining was performed overnight at 4 °C in the dark.

For indirect labelling with anti-gp120 antibodies, viruses were incubated with 0.4 µg/mL of unlabelled primary mAbs overnight at 4 °C, followed by a three-hour incubation with 0.2 µg/mL of a rat anti-human PE-labelled secondary antibody (BioLegend, San Diego, CA, USA, Cat# 410708). The optimal concentration of secondary antibody for staining with the highest ratio of specific signal to minimal levels of background was determined through titration of both the BioLegend and eBioscience antibodies (San Diego, CA, USA) Cat# 12-4998-82). For the select few antibodies that were produced in mouse hybridomas, a goat anti-mouse PE-labelled secondary antibody (Invitrogen, Waltham, MA, USA, Cat# A10543) was used. For staining experiments with soluble CD4 (NIH ARP-7356), virus was incubated with 10 µg/mL final of soluble CD4 for 20 min before staining was completed as described above. For direct labelling protocols with PG9 and PGT145, viruses were stained overnight at 4 °C with 0.4 µg/mL of the anti-gp120 antibodies conjugated with R-phycoerythrin (PE) in-house. 

After staining, viruses were fixed in a final concentration of 2% PFA for 20 min. Prior to acquisition on the cytometer, all samples underwent an additional dilution with PBS (1:500) in order to minimize coincidence. BD Quantibrite PE beads (Franklin Lakes, NJ, USA; Cat#340495, lot 91367) and NIST-traceable size standards (Thermo Fisher Scientific, Waltham, MA, USA) were used for fluorescence and light scattering calibration, respectively, as described before [48]. Calibration was performed using FCM_PASS_ software version 4.2.4 (https://nano.ccr.cancer.gov/fcmpass, accessed on 3 June 2024; Appendix A) as previously described [42,57]. All data were analyzed using FlowJo software version 10.7.1. (Ashland, OR, USA). PE MESF statistics were generated from gates set on the virus population (as described in the text) using FlowJo. 

### 2.4. Plate-Based Virus Immunocapture Assay

Sterile tissue culture plates were coated overnight at 4 °C with 5 μg/mL of anti-Env antibodies 246-D, PG9, PGT126, or with an isotype control antibody (Invitrogen Cat# 02-7102). The following day, wells were washed three times with PBS before being blocked with 1% BSA in PBS at room temperature for 1 h. Following blocking, the wells were washed as before, and undiluted virus stocks were added to the wells for overnight capture at 4 °C. The captured virus was lysed with 0.5% Triton X-100, followed by p24 quantification by AlphaLISA. Data analysis was performed using Prism 9.5.0 (GraphPad, San Diego, CA, USA). To validate the results of the plate-based virus assay, immunomagnetic bead-based virion capture assays (in suspension) were performed as described previously [47,48].

### 2.5. p24 AlphaLISA

The quantification of HIV-1 p24 capsid protein was performed in lysates of captured virus with the AlphaLISA p24 detection kit following the manufacturer’s (PerkinElmer, Waltham, MA, USA) instructions. Absorbance readings were performed on a Synergy NEO 2 multimode plate reader (BioTek, Winooski, VT, USA) using Gen 5 software (v. 3.08).

### 2.6. Neutralization Assay

For neutralization assays, PG9 and PGT126, prepared at the concentrations indicated, were pre-incubated with each virus for 1 h at 37 °C, followed by a 48 h incubation with 10,000 TZM-bl reporter cells. Virus neutralization was monitored by adding Britelite plus reagent (PerkinElmer) to the cells, transferring cell lysates to PerkinElmer 96 well OptiPlates (Cat#6005299) and measuring luminescence in relative light units using the Synergy NEO 2 multimode plate reader (BioTek). All the samples were tested in triplicate wells. 

## 3. Results

### 3.1. Using Flow Virometry to Evaluate the Staining of a Diverse Repertoire of Epitopes on HIV Env

While we have previously used our flow virometry protocols to study cellular proteins on the surface of HIV directly in cell culture supernatants [47,48,52], we embarked on this study because these protocols had not yet been applied to the viral envelope glycoprotein, HIV Env. Thus, we sought to test whether we could stain diverse sites on the HIV Env reliably and sensitively using a large panel of publicly available monoclonal antibodies (mAbs). With this in mind, we chose to evaluate the 85 mAbs available in the NIH HIV Reagent Program’s catalog, a significant portion of which are well described, patient-derived mAbs. To begin, the CXCR4-tropic, laboratory-adapted strain HIV_IIIB_ was produced from the H9 T cell line as a model virus. Prior work from our group has shown these virus populations to be monodisperse and homogenous, whereas viruses produced in primary cells can be more heterogenous, particularly when analyzed by flow virometry [48]. Since none of the anti-Env mAbs from the HIV Reagent Program were available as PE-conjugates, we used indirect staining with a PE secondary Ab for Env staining, since our flow virometry protocols were previously optimized for PE detection. To initially screen the panel of antibodies in a high-throughput manner, we chose to test all primary antibodies at a concentration of 0.4 µg/mL since this concentration has been used reliably by us and others in the past [47,48,49]. To begin, we ran PBS on the instrument and compared it to unstained virus samples. As expected, a notable virus population was detectable by light scatter, as seen in the bottom gate in Figure 1A. To establish our level of fluorescence background, we tested two different human isotype control mAbs and saw negligible levels of non-specific staining with both antibodies (Figure 1A). Using these controls, we set an upper gate that displays positive virus staining, whereas the lower gate was set to encompass the background fluorescence.

After establishing the validity of these controls, the full panel stain with 85 diverse anti-Env mAbs was performed, with selected stains in Figure 1B and all quantitative results shown in Table 1. Low to moderate staining was seen on a wide range of epitopes from well-defined Env trimer domains, including variable loops 1 and 2 (V1V2), the V3 loop base, the transmembrane domain gp41, and the CD4 binding site (CD4bs). Out of the 85 mAbs tested, 64% (54/85) showed staining that was above the level of non-specific staining seen with the isotype controls (Figure 1A). Select clones which demonstrated some of the top staining from each class of mAbs are shown in Figure 1B, with PE fluorescence reported in calibrated units of PE fluorescence, termed molecules of equivalent soluble fluorophore (PE MESF). Notably, bNAbs targeting variable loop 3, such as PGT126 and PGT128, yielded the highest PE MESF values (~11 MESF). Despite the fact that CD4bs mAbs like N6 and VRC01 are particularly potent and neutralizing [24], these mAbs showed relatively low staining, just barely exceeding levels of staining seen with the isotype controls. Trimer-preferring V1V2 mAbs, such as PG9 and PG16, showed more appreciable staining (~3–7 MESF), whereas mAbs targeting the transmembrane domains in gp41 showed modest levels of staining (<3 MESF). Interestingly, many mAbs known to potently neutralize HIV, such as PGT145, showed very low levels of staining in our flow virometry assays, levels that were close to the threshold of background fluorescence. However, it should be noted that this low level of detection could also be due in part to the lower sensitivity seen with indirect staining protocols and the resolution of our cytometer. It should also be noted that we tested all of the publicly available anti-Env Abs. Some of the low levels of staining may simply be because these antibodies were not specifically reactive for the IIIB Env sequence. As a control for non-specific staining, we reported negligible levels of staining when the same panel of antibodies were stained on cell culture supernatants from matched, uninfected H9 T cells (Appendix A).

### 3.2. Comparing Env Staining across Different Cellular Models of Virus Production

After demonstrating that numerous mAbs from our panel could stain Env, we sought to assess how viruses generated through different cellular models of virus production may impact Env staining. For this purpose, we chose one representative antibody targeting V1V2, the V3 loop, and gp41 for further experiments. Using the neutralizing mAbs PG9 (V1V2) [58], PGT126 (V3) [59], and the non-neutralizing Ab 246-D (gp41 apex) [60], we stained four different viruses of both coreceptor usages generated in diverse CD4^+^ T cell lines, including H9, Jurkat, and PM1 cells. As expected, for all of the isolates tested, negligible levels of staining were seen with the isotype controls on virus-containing supernatants (Figure 2A, top row). Staining with the bNAbs PGT126 and PG9, which in general prefer closed trimer conformations [13,61], showed high staining across all viral isolates, except for the Jurkat IIIB isolate, which showed modest levels of staining. Interestingly, 246-D, an anti-gp41 Ab [62], which targets a more open Env conformation, also showed modest levels of staining across all of the T cell line isolates, with the highest intensity staining seen on Jurkat IIIB and PM1 BaL isolates. Of note, PGT126, and PG9 also showed high degrees of staining on particles outside of the virus-specific gate, suggesting that extracellular vesicles (EV) may also be staining with these bNAbs, as previously reported [29]. Labelling with 246-D yielded lower levels of EV staining, indicating this epitope may be less abundant on EVs. It should also be noted that some EVs can overlap in scatter profiles and appear within our virus-specific gate, as observed when assaying matched, uninfected cell culture supernatants, where a lower number of non-discrete, heterogeneous, EV particles are observed in the lower gate (Appendix A). However, when performing anti-Env staining on these EV control supernatants, we did not observe any appreciable positive staining, indicating that while some EVs may be present in the virus-specific gates, these EVs are likely not contributing to the positive Env staining detected in the upper gate for our virus samples. Of note, it is possible that additional EVs produced by infected cells may carry HIV Env on their surface, and those particles may contribute to some of the positive anti-Env staining reported in our virus-specific gate.

Since viruses produced through transfection of HIV constructs are routinely used to assess virus neutralization efficacy [63,64,65], we next assessed how similar the staining profiles of viruses produced in HEK293T cells were to viruses produced in T cell lines. For this purpose, the HEK293T-derived NL4-3 and NL4-3-BaL full-length, infectious molecular clones (IMC) were stained with the same mAbs. While both isolates displayed positive staining, the NL4-3 isolate with the BaL envelope (NL4-3-BaL) showed superior staining for all mAbs, with the wild-type NL4-3 isolate only showing a modest level of staining above the background (Figure 2B; left panels). Surprisingly, in contrast to what was seen with viruses produced in T cell lines, 246-D yielded the highest levels of staining on both IMCs, with a clear shift in the virus population (Figure 2B). Staining outside of the virus gate was also present, but at a lower intensity than seen with T cell line-derived virions. Importantly, while the levels of staining seen from HEK293T-derived viruses yielded PE MESF values in similar ranges to the Jurkat IIIB (~3–8 MESF), these values were two-fold lower than the result from PGT126 stains on IIIB produced in the H9 or PM1 cell line (Figure 2A). This suggests that notable differences in staining intensity can occur when virus isolates are produced through different protocols and in different cell lines. Additionally, this statement holds true when comparing the data in Figure 2A amongst CD4^+^ T cell lines, where the identical virus isolates produced differential staining in different cell types, as observed when comparing staining on Jurkat-derived versus H9-derived IIIB virus.

Finally, since viruses produced in primary cells most closely resemble those found in vivo, we tested the HIV_BaL_ isolate produced in two different PBMC donors. When stained, both PBMC isolates displayed trends that were similar to those seen in T cell line-derived viruses (Figure 2B; right panels). However, for PBMC viruses, PG9 staining showed slightly higher MESF values than PGT126 staining, which was in contrast to what was seen with the same isolate propagated in the PM1 T cell line in Figure 2A. Since PG9 is a glycan-dependent antibody [66], the differences in staining may be partially attributable to differences in glycosylation across the array of cell lines/types used for virus production. Notably, while the viruses from PBMC displayed less homogenous virus populations than what was seen in viruses from cell lines, this was in line with what we have reported previously [48]. Strikingly, when we stained pseudoviruses (PV) bearing a BaL.01 envelope, Env staining was undetectable using the same protocol that was applied to the other viruses, despite the pseudoviruses being able to be neutralized and captured by the bNAbs PGT126 and PG9 (Appendix A). 

While Env on virions produced in HEK293T cells and PBMC have been reported to share highly similar glycan processing sites [67], the complex glycans could still differ in branching and terminal sialic acids, which could affect staining [67]. Since our lab routinely transfects HEK293T cells to produce viruses with CD162 on their surface, to validate that our PV model could be stained effectively, we produced PVs in the same HEK293T cells as used before with BaL.01, but this time we expressed the cellular protein CD162 on the surface. After staining these virions with anti-CD162 antibody, we were able to see high levels of CD162 staining on PVs (Appendix A), indicating that robust staining can be seen on PVs in flow virometry, depending on the antigen targeted. Since high-abundance cellular proteins can readily be detected on PV, it is possible that the number of HIV Env trimers on these PV preparations falls below the limit of detection of our cytometer. Indeed, prior reports indicate that PVs typically contain 6–20 Env trimers, but that most viral strains only need 1–3 trimers to complete infection [68,69]. This could in part explain why our viruses are infectious despite showing undetectable Env staining (Appendix A). 

### 3.3. Assessing the Performance of Antibodies in Virion Capture Assays and Virus Neutralization

After noting that certain broadly neutralizing mAbs tested in the panel screen did not yield high levels of staining in flow virometry assays (Figure 1B), we next sought to assess how mAbs that perform well in flow virometry compare to classical virology assays. To begin, the same viruses from above were tested in a plate-based virus immunocapture assay that we and others have previously used [48,70,71,72] to evaluate antibody-mediated virion capture. For this purpose, we continued to use the anti-Env mAbs PGT126, PG9, and 246-D since they provided reliable staining across all of the viruses produced in T cell lines. For viruses produced in T cell lines, PG9 and 246-D yielded robust and moderate levels of capture, respectively, as assessed by the readout of captured virions by quantification of p24 (capsid) AlphaLISA. In contrast to what was seen in flow virometry, PGT126 showed low levels of capture for all viruses, except for the Jurkat IIIB isolate (Figure 3A). Notably, although PGT126 yielded the highest levels of staining in flow virometry for the other three T cell line-derived viruses (PM1 BaL, PM1 IIIB, H9 IIIB), this trend was not reflected in virus capture. To ensure that this was not an artefact of our plate-based capture assay, this experiment was repeated using bead-based capture, in which mAbs are immobilized on Protein G dynabeads for viral capture in suspension, and these parallel experiments yielded similar results (Appendix A). 

Assessments of PBMC-derived BaL using plate-based capture showed similar results to the H9 and PM1 IIIB isolates (Figure 3B), and the same trend was also seen in IMC viruses (Figure 3C). While the discrepancy in levels of PGT126-mediated virus capture versus staining in FV was surprising, we did anticipate seeing some differences in the two assays given that they use different endpoint readouts. Notably, a similar difference in levels of the virion-incorporated protein CD81 was previously reported by our group [47] when comparing results from virus capture assays versus flow virometry staining. Since both CD81 and PGT126 mAbs demonstrate notable levels of extracellular vesicle staining, these discrepant data could suggest that extracellular vesicles may be contributing differently to the results acquired in these two assays. 

Based on the seemingly different Env binding efficiencies of antibodies in FV and immunocapture, we sought to compare how these mAbs performed with our virus stocks in another commonly used antibody-based assay, virus neutralization. We performed neutralization assays comparing the activity of two broadly neutralizing antibodies against HIV-1, PGT126 and PG9, while 246-D was not tested here, as it is known to be non-neutralizing [73]. Since the neutralization profile of PGT126 and PG9 is well known, we narrowed our analysis to three viruses from T cell lines, two from PBMC, and one from HEK293T. Unsurprisingly, PG9 and PGT126 demonstrated a potent neutralization of all of the viruses produced in T cell lines, PBMC, and HEK293T (Figure 4). These results highlight that the differences seen in virus capture and staining using PG9 and PGT126 are not related to their neutralization potential, as we observed similar neutralization profiles with these mAbs, despite highly divergent capture efficiencies (Figure 3). Neutralization assays were also performed on pseudovirus particles produced through transfection in HEK293T cells (Appendix A), which showed potent neutralization sensitivity despite no detectable staining with flow virometry (Appendix A). Thus, pseudovirus particles can infect HIV target cells, presumably via the presence of Env on their surface, yet unexpectedly, anti-Env staining cannot be detected on pseudoviruses via the same flow virometry assays used herein that work well on other virus models.

### 3.4. Using Flow Virometry to Evaluate the Effect of Soluble CD4 on HIV Envelope Conformation

To determine whether flow virometry can reliably be used to detect conformational changes in the envelope glycoprotein, we sought to determine whether differences could be observed when viruses were stained in the presence and absence of soluble CD4 (sCD4), since CD4 induces changes in Env conformation during the process of viral entry [74]. For these studies, we limited our analyses to HIV_IIIB_ produced in the H9 CD4^+^ T cell line since these virus stocks yielded the highest quantitative levels of Env staining (Figure 2), allowing us to better resolve subtle differences in conformation. To begin, the virus was pre-incubated with sCD4 for 20 min, followed by incubation with mAbs specific to the CD4bs (b12), the coreceptor binding site (E51) [75,76], or gp41 (50–69) [77], in anticipation that these mAbs would reveal differences in Env conformations. In parallel, we also tested PG9, PGT126 and 246-D to allow comparison to prior datasets. 

The antibody E51, which targets a CD4-inducible (CD4i) epitope, showed some appreciable staining above background in the absence of sCD4, and this staining was seemingly unchanged by the addition of sCD4 (Figure 5A). However, the addition of sCD4 resulted in reduced b12 staining, as expected since b12 targets the CD4 binding site and should be occluded in the presence of sCD4 (Figure 5A,B). Similarly, both PG9 and PGT126, which preferentially target the closed form of the trimer [13,61], showed a marked reduction in staining when sCD4 was present, as evidenced both from the dot plots and quantitative data (Figure 5A,B). Contrarily, 246-D and 50–69, which target epitopes on the transmembrane gp41 of Env, showed a marked increase in staining after sCD4 addition promoted the open conformation of the trimer. These results were in line with previous studies which showed sCD4 can enhance the accessibility of gp41 epitopes [78]. Taken together, these results demonstrate the utility of flow virometry to assess different Env conformations.

### 3.5. Staining HIV Env with Direct Labelling to Improve Detection Sensitivity

Since antibodies conjugated with the fluorophore PE are expected to have one fluor molecule per individual antibody [79], and because PE fluorescence can be quantitated using well-established reference materials [80,81], staining with PE-labelled antibodies in flow virometry can provide quantitative estimates of protein content on virus particles. While we have previously used direct staining protocols to generate quantitative estimates of human proteins on the surface of viruses [47,54], we wanted to apply our technique for enumerating the HIV envelope protein in this study. However, since commercial preparations of PE-labelled anti-Env antibodies are not readily available, we conjugated anti-Env mAbs in-house to generate PE-labelled antibodies for use in quantitative flow virometry. Furthermore, since indirect staining has proven to be less sensitive than direct staining in our experience, we anticipated that direct labelling of Env would also help increase the quantitative levels of detectable Env staining. To see if we could improve the sensitivity of mAbs that yielded both low and high levels of labelling in indirect staining (Figure 1), we selected the mAbs PGT145 and PG9, respectively. Here, we chose to once again stain the HIV_IIIB_ virus produced in H9 CD4^+^ T cells for consistency of data comparisons. When performing direct staining with the conjugated mAbs, a modest visible increase in PE fluorescence was present in direct staining dot plots (bottom row) compared to indirect labelling plots (top row) for both antibody clones (Figure 6). This was observed as the shifting of the dense, homogenous virus populations upward, both into the upper gate and towards the top of the bottom gate in the direct staining plots. This increase in staining was also quantifiable, with an increase in PE MESF from 13 to 17 for the PG9 stain, and a more modest increase from 4 PE MESF to 5 PE MESF for the PGT145 stain, when comparing indirect to direct staining, respectively. However, levels of background fluorescence at the instrument threshold also increased slightly with the conjugated antibody. Nevertheless, these data indicate that direct staining can improve signal detection for HIV Env, as we observed increases in the PE MESF values for both PG9 and PGT145 staining when directly labelled antibodies were used. As expected, PG9 labelling demonstrated higher levels of staining than PGT145 in both indirect and direct labelling methods, and negligible levels of staining were observed on uninfected H9 T cell culture supernatants (Appendix A). Notably, while both methods of staining can provide quantitative PE MESF staining values when used with fluorescence reference beads and data calibration, it is expected that direct staining provides quantitative values that provide better estimates of the number of proteins on the virion surface. This is due to the fact that each primary antibody in direct labelling is likely associated with one molecule of PE, due to the expected fluor/protein ratio of 1 [79]. Based on this, the number of individual antibodies bound to each virion can be inferred, which provides an estimate of the number of proteins present [80,81], acknowledging that this estimate could vary by a factor of two, due to the bivalent nature of mAbs. However, in indirect staining methods, several secondary antibodies could be bound to one primary antibody, which can further complicate quantitative estimates. Despite this possibility, we did not observe any enhanced fluorescent signal of indirect staining compared to direct staining methods (Figure 6). Moreover, while optimizing our indirect labelling protocols for this study, we noted that the clone and lot number of the secondary antibody could significantly influence the staining intensity of some primary antibodies and the levels of non-specific, background fluorescence (Appendix A). Although high levels of signal are important for labelling low-abundance antigens like Env, we opted to use an antibody with lower levels of signal that yielded minimal levels of background fluorescence to ensure that our Env staining was specific and not due to antibody noise.

## 4. Discussion

To develop effective vaccines and other antibody-based strategies to combat HIV infection, it is critical to establish a thorough understanding of how different antibodies bind diverse strains of circulating virus. Indeed, while many antibody-based techniques have helped generate important knowledge of the HIV envelope trimer, the use of new tools will further drive our ability to better understand Env structure and function. Flow virometry is an underutilized tool that can continue to help in understanding more about virion heterogeneity, Env conformations, and how Env interacts with antibodies. For instance, we observed clear differences in how the antibodies PGT126, PG9, and 246-D performed across capture, neutralization, and flow virometry, which may be overlooked when relying solely on classical virology techniques. Indeed, one distinct advantage of flow virometry is the ability to visualize heterogeneity in virus populations on dot plots, which can provide additional information for downstream applications targeting HIV Env. Similarly, flow virometry dot plots also allow for the visualization of extracellular vesicles within virus samples, which may help broaden our understanding of how vesicles contribute to antibody interactions with Env, as carried out previously [29].

When studying model viruses for the purpose of designing effective vaccines against HIV, many considerations remain, including ensuring physiological levels of trimer abundance and N-linked glycosylation patterns [82]. In this study, we stained viruses from different cellular models (HEK293T, T cell lines, PBMC) and observed considerable differences in the antibody staining profiles. These findings highlight that when testing therapeutics and vaccines, special consideration must be given when selecting model viruses to ensure the data acquired can be relevant and useful for understanding circulating viruses in people living with HIV. For instance, while pseudoviruses are commonly used in studies which assess antibody neutralization [83,84,85], our data indicate that pseudovirus Env was substantially less detectable by our flow virometry protocols than viruses produced through transfection with full-length virus clones. While prior reports have suggested that levels of Env and trimer glycosylation sites are similar across viruses produced in HEK293T cells, T cell lines, and PBMC [67,68], our data indicate that these comparisons necessitate further study. Nevertheless, flow virometry provides a simple methodology to better understand how the structure and antigenicity of the trimer may change when viruses are produced in different cell types. Indeed, flow virometry could also prove useful to characterize circulating strains of viruses directly in biological samples from people living with HIV, which is a subject of ongoing work in the laboratory.

Our results indicate that several broadly neutralizing and non-neutralizing anti-Env antibodies, both of which are influenced by the trimer conformation, can provide high levels of staining by flow virometry. Although our typical flow virometry protocols utilize directly conjugated antibodies, in this manuscript we used indirect staining to label Env due to a lack of commercially available PE-labelled anti-Env mAbs. Our results demonstrated that even different lots of the same secondary antibody could have major differences in staining efficacy when using the same primary antibody. For this reason, a variety of secondary antibody preparations should be tested with indirect staining protocols to ensure experimental reproducibility and appropriate controls for quantitative applications. Similarly, while antibody titration is always beneficial, maximizing specific signal while reducing nonspecific binding is especially important in flow virometry protocols, since no wash steps are conducted. For this reason, ensuring unbound fluorophores are removed and/or controlled for in antibody preparations is essential for reliable interpretation of staining results. Overall, the use of direct staining protocols remains a best practice to remove potential errors in staining quantifications due to interactions between the primary and secondary antibodies.

While we were able to detect staining for the majority of antibodies tested using the indirect labelling protocol here, it is possible that improved staining sensitivity would be achieved with labelled Fab fragments. Indeed, it has been previously reported that for CD4i mAbs, the size of the neutralizing agent can be inversely correlated with its ability to neutralize [86]. Thus, steric hindrance is a major consideration in these types of antibody assays for small particles, particularly when using dual labelling approaches. Although we report quantitative estimates of PE for all data within the manuscript, it is highly likely that MESF values yield more accurate estimates of protein number on viruses when antigens are in high abundance. Since many of the antibody stains shown here were near the level of instrument background noise, the quantitative estimates of Env number per particle are likely underestimated. However, the currently reported PE MESF for Abs such as PGT126 and PG9 values are more consistent with prior reports quantifying 8–14 Env per virion [9,87]. Reducing instrument noise remains a top priority in flow virometry techniques, as even in the absence of any virus, the electronics, fluidics, and optical system within the cytometer can contribute a significant signal that is present on each plot generated by the instrument [88]. The use of careful gating based on reagent controls can help identify where virus-specific signals fall. 

Importantly, our results show that conformational changes in the HIV trimer were readily detected in flow virometry when soluble CD4 was present. Of note, the no-wash flow virometry protocol used herein allows for the study of viruses in their native state without additional factors (e.g., coupling to magnetic beads, shedding from ultracentrifugation, harsh fixatives in sample processing), which could bias Env evaluation. Indeed, while much has been uncovered about HIV Env biology through the use of stabilized trimeric mimics of the HIV Env (SOSIP trimers [89,90,91]), these recombinant proteins may not fully recapitulate native Env trimers as presented on infectious viral particles. Additionally, soluble SOSIP Env can show striking differences in glycosylation, and this may impact antibody recognition of Env [67]. Furthermore, performing flow virometry-based Env characterization on a broader range of diverse viral isolates, including transmitter-founder strains, is an important future direction to determine how differences in Env sequences and post-translational modifications are detected with our unique assays used herein.

Finally, while others have studied the Env trimer on virions using flow virometry, the methods reported herein provide an advantage by removing the necessity for fluorescently tagged virions [56] or coupling to magnetic nanoparticles [28,29] to enable detection of virions on conventional cytometers. While previous studies that evaluated Env on virions with flow virometry techniques have all used different methodological approaches [32,55,56], most studies have consistently reported successful targeting with the anti-Env antibodies PG9, PG16, 2G12, and VRC01. Interestingly, while PG9 and PG16 also showed robust staining in our unique flow virometry assays, 2G12 and VRC01 yielded low levels of staining. At this time, we are uncertain as to why 2G12 and VRC01 show poor targeting in flow virometry assays but acknowledge that this could in part be influenced by the secondary antibody used in our indirect staining protocol. In the future, it would be useful to perform similar anti-Env stains on viruses displaying Env mutants with distinct conformations to discern the nuance of our assay performance and its correlation with Env confirmations. Importantly, with emerging developments in nanoparticle-specific flow cytometry instrumentation and reagents, we anticipate that flow virometry will be able to provide more sensitive readings of low-abundance antigens and make it difficult to target epitopes on viral surfaces in the near future. Furthermore, as sorting technologies continue to improve, purifying viruses from clinical samples with anti-Env mAbs, like those used in this study, should become more feasible than prior protocols [55], enabling additional single particle analyses on virus subpopulations of interest.

## Figures and Tables

**Figure 1 viruses-16-00935-f001:**
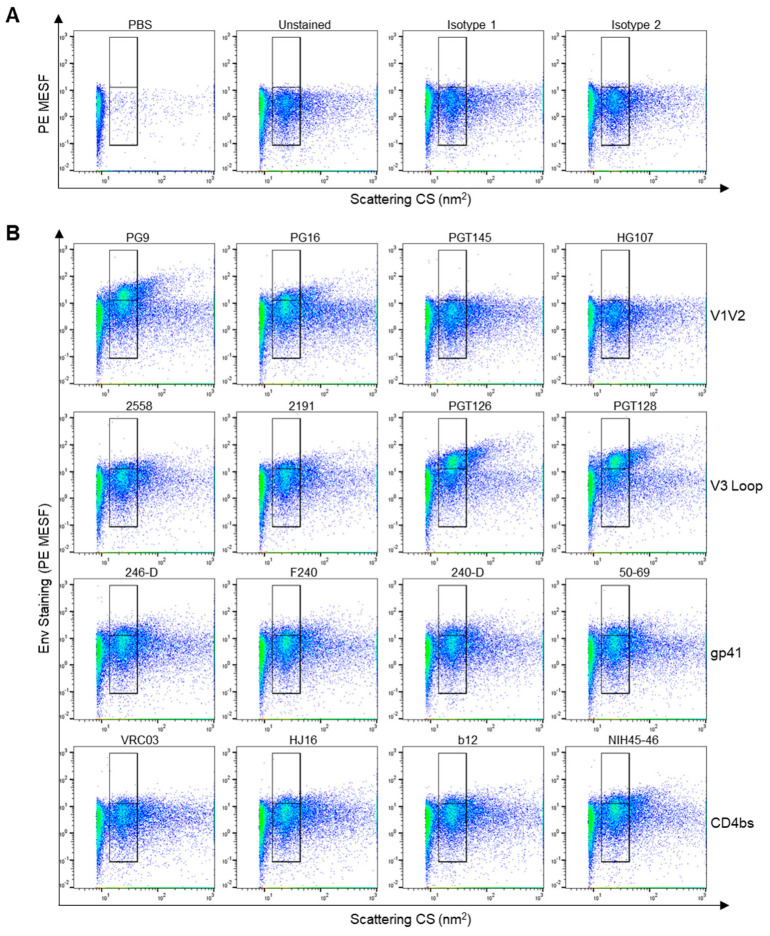
Anti-Env staining on the HIV_IIIB_ isolate produced in H9 CD4^+^ T cells. (**A**) Dot plots displaying flow virometry control stains, including PBS alone, unstained virus, and virus stained with different anti-human isotype control antibodies. Gates are set on the scattering profile (x-axis) of the HIV virus. Positive staining is shown in the upper gate, while background levels are within the lower gate, as determined with isotype controls. (**B**) Selected antibody staining from quantitative data reported in Table 1, with mAbs targeting the variable loops 1 and 2 (V1V2), V3 loop, gp41 or the CD4bs shown across each row. Representative plots from three independent experiments are shown.

**Figure 2 viruses-16-00935-f002:**
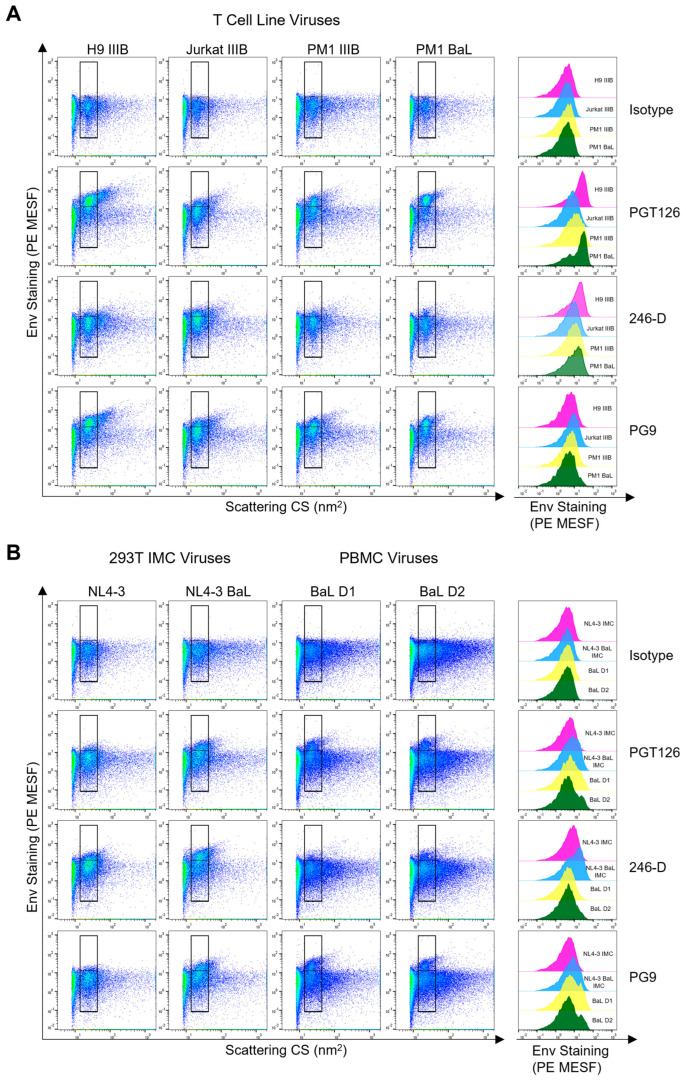
Comparing Env staining across different cellular models of virus production. (**A**) Dot plots displaying anti-Env antibody staining (PGT126, 246-D, PG9) of viral isolates (IIIB or BaL, as indicated) produced in the H9, Jurkat, or PM1 cell lines. Positive staining is shown in the upper gate, while background levels as assessed by the isotype control are shown in the lower gate. Histogram overlays displaying the range of Env staining for each virus are shown to the right of the dot plots. The levels of PE-fluorescence represented on histograms are generated from the total virus staining (i.e., spanning the upper and lower gates). (**B**) In the left two panels, dot plots display anti-Env antibody staining of viruses (NL4-3 and NL4-3 BaL) produced through transfection of infectious molecular clones (IMC) in HEK293T cells. In the two rightmost panels, plots display antibody staining of the HIV_BaL_ isolate produced in PBMC from two different donors (D1 and D2). Representative plots are shown from at least two independent experiments.

**Figure 3 viruses-16-00935-f003:**
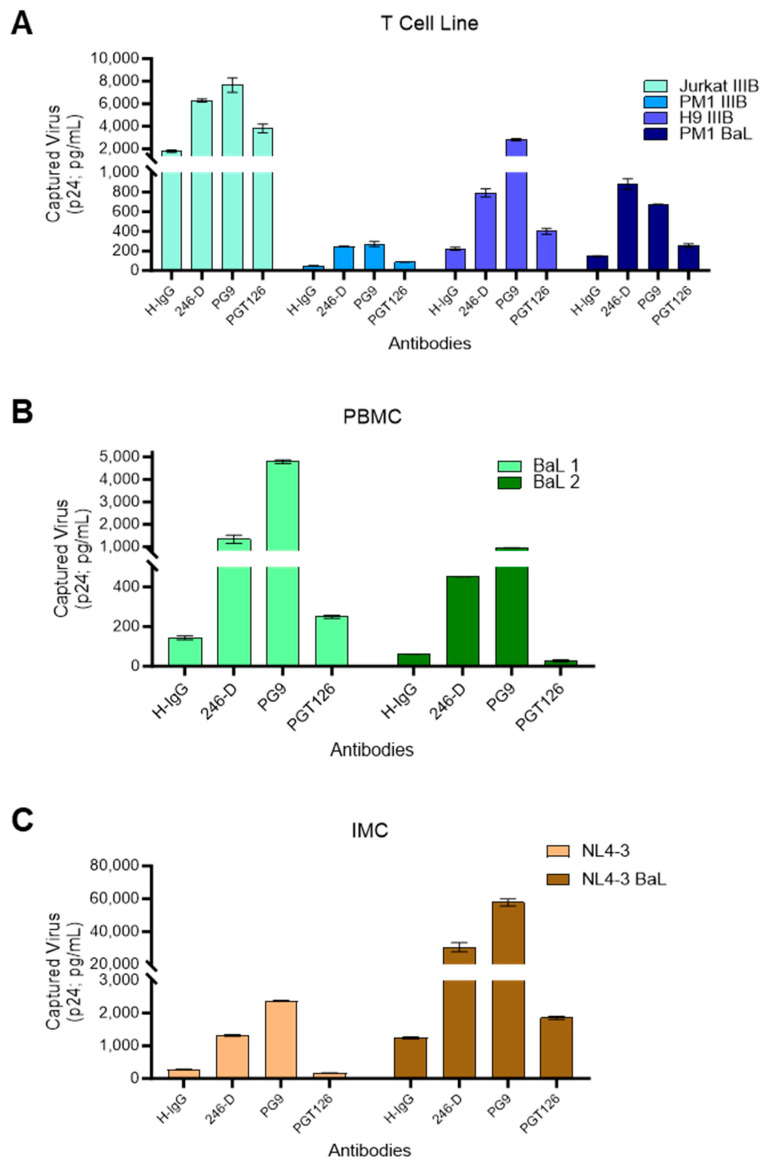
Comparison of virus immunocapture with anti-Env antibodies across different virus model systems. Plate-based virion immunocapture assays were performed with wells coated with the antibodies PGT126, 246-D, PG9, or an isotype control. Captured viruses were lysed and HIV-1 p24 Gag was quantified using p24 AlphaLISA as an indicator of the amount of virus capture. (**A**) Viruses (IIIB and BaL) produced in T cell lines (PM1, Jurkat, H9) were added to the wells at a normalized concentration of input virus (50 ng/mL of viral p24). (**B**) The HIV_BaL_ isolate propagated in two different PBMC donors (BaL 1 and BaL 2), and (**C**) viruses produced from the transfection of infectious molecular clones (IMC; NL4-3 BaL and NL4-3) in HEK293T cells were tested at their undiluted titers. Results represent the mean ± SD of duplicate wells and are representative of three independent experiments.

**Figure 4 viruses-16-00935-f004:**
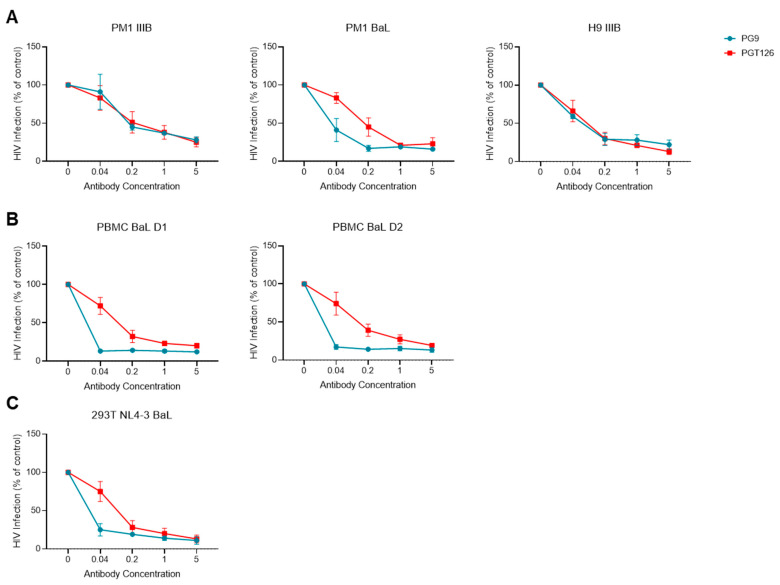
Validating the neutralization profile of mAbs PGT126 and PG9 against different viruses. (**A**) Neutralization sensitivities of IIIB and BaL viruses grown in the PM1 and H9 CD4^+^ T cell lines using PG9 (blue) and PGT126 (red) mAbs. (**B**) Neutralization of HIV BaL produced in two different PBMC donors (D1 and D2). (**C**) Neutralization of virus produced through the transfection of HEK293T cells with the NL4-3-BaL infectious molecular clone. All neutralization tests were performed using the TZM-bl reporter cell assay, with the infection of control samples (in absence of mAbs) set at 100%. Data are representative of two independent experiments tested with triplicate wells.

**Figure 5 viruses-16-00935-f005:**
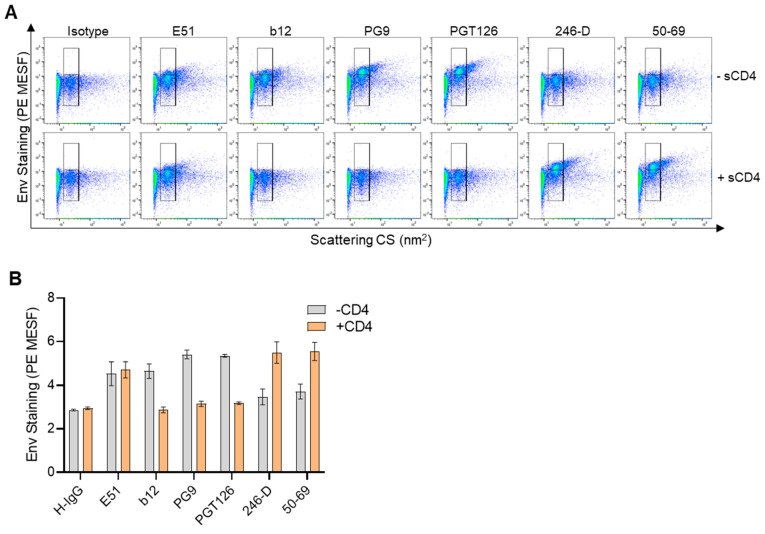
Staining HIV_IIIB_ in the presence or absence of soluble CD4. (**A**) Dot plots depicting flow virometry staining of HIV_IIIB_ in the presence (+CD4) or absence (−sCD4) of soluble CD4. Positive staining is shown in the upper gate, while background levels fall within the lower gate. Plots shown are representative from three independent experiments. (**B**) Quantified staining data showing the mean ± SD of three replicates as in (A), generated from the individual median PE MESF values derived from all events within both virus gates (upper and lower) of each replicate stain.

**Figure 6 viruses-16-00935-f006:**
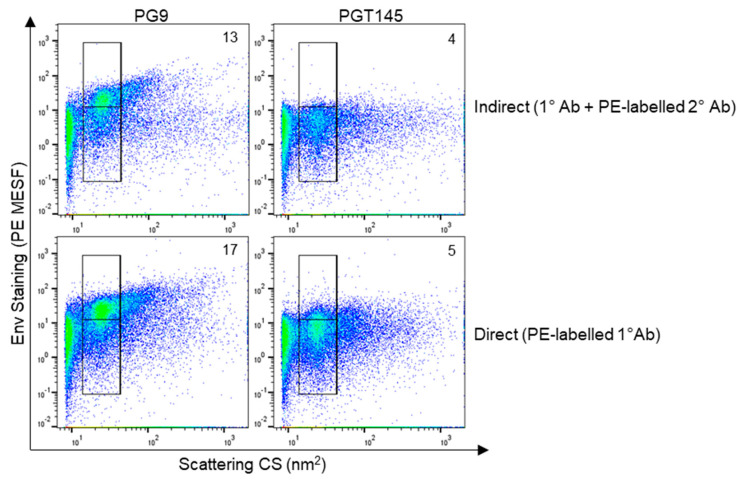
Comparing direct and indirect staining methods in flow virometry to label HIV Env. Indirect staining of HIV_IIIB_ propagated in H9 CD4^+^ T cells with unlabeled primary anti-gp120 antibodies (PGT145 and PG9), and a PE-conjugated secondary antibody (**top row**). Direct staining of the same HIV_IIIB_ virus stocks using PE-labelled PGT145 and PG9 antibodies (**bottom row**). Positive staining is shown in the upper gate, while background levels fall within the lower gate. Median PE MESF values that were generated from all virus events within the merged upper and lower gates are enumerated on the plots.

**Table 1 viruses-16-00935-t001:** Quantitative staining data from an anti-Env antibody panel performed on HIV_IIIB_ produced in H9 CD4^+^ T cells. Quantitative data represent the PE MESF statistics from a gate encompassing all virus events (i.e., the sum of the upper and lower gates in Figure 1) after subtraction of background fluorescence, as measured by two isotype controls. Stains with high intensity are indicated by a darker shade of red, with low intensity stains in yellow, and absent stains unreported and shaded white. Data represent the mean and standard deviation of the median PE MESF values from two independent experiments. * N6/PGDM1400x10E8 is tri-specific antibody.

HIV Env Trimer Domain	Antibody	ARP Catalog #	Mean (PE MESF)	SD
**V3 glycan supersite**	257-D IV	1510	0.08	0.15
	F425 B4e8	7626	-	-
	39F	11437	-	-
	PGT126	12344	11.27	0.19
	10065D	13426	-	-
	2557	13429	0.60	0.27
	2558	13432	1.49	0.28
	2191	11682	1.98	0.34
	10074	12477	1.42	0.28
	447-52D	4030	0.65	0.10
	F425 B4a1	7625	0.01	0.06
	3074	12040	-	-
	2G12	1476	0.62	0.12
	268-D IV	1511	-	-
	3869	12039	-	-
	PGT128	13352	10.92	0.64
	2219	11683	-	-
	PGT121	12343	0.74	0.16
	5F7	2533	0.57	0.20
**gp160**	902	522	1.16	0.34
	Chessie 6	810	0.20	0.08
	ID6	2343	0.46	0.14
**gp120**	IgG YZ23	12047	0.05	0.14
**Constant region 1**	A32	11438	-	-
	654-30D	7369	-	-
	CH38	12548	-	-
	CH57	12549	-	-
	CH90	12552	-	-
	A32-AAA	12558	0.02	0.09
	16H3	12559	0.17	0.07
	3B3	12560	0.16	0.08
	Chessie 13-39.1	1209	0.09	0.06
**Constant region 5**	670-30D	7370	0.06	0.06
**CD4bs**	VRC01	12033	0.21	0.10
	3BNC117	12474	0.27	0.07
	VRC-CH31	12565	-	-
	CH106	12566	0.14	0.08
	F105	857	0.02	0.09
	* N6/PGDM1400x10E8	13390	2.53	0.35
	N6	12968	0.22	0.19
	VRC03	12032	1.19	0.22
	NIH45-46 G54W	12174	1.85	0.57
	IgG1 b12	2640	1.47	0.07
	HJ16	12138	1.52	0.09
**CD4i**	E51	11439	2.76	0.14
	scFv m9	11710	0.05	0.08
	F425 A1g8	7624	1.07	0.37
	17b	4091	0.81	0.14
	48d	1756	0.95	0.23
**gp120 interface**	35022	12586	0.07	0.08
**MPER**	Z13e1	11557	-	-
	10E8	12294	-	-
	7H6	12295	-	-
	10E8v4	12865	-	-
	2F5	1475	-	-
	4E10	10091	-	-
**V1V2**	CH58	12550	-	-
	CH59	12551	-	-
	HG107	12553	-	-
	HG120	12554	-	-
	CH01	12561	-	-
	CH01	12562	-	-
	CH03	12563	-	-
	CH04	12564	-	-
	PG9	12149	7.62	0.16
	PG16	12150	2.94	0.24
	2909	12141	-	-
	PGT145	12703	0.08	0.07
	697-30D	7371	-	-
	Ibalizumab (PG9-iMab)	12633	0.40	0.07
**gp41**	246-D	1245	1.95	0.19
	7B2	12556	1.02	0.32
	7B2-AAA	12557	1.87	0.07
	240-D	1242	1.74	0.31
	T32	11391	0.47	0.12
	NC	11482	0.08	0.06
	5F3	6882	0.01	0.18
	F240	7623	1.97	0.12
	D50	11393	0.11	0.08
	1577	1172	0.08	0.06
	Chessie 8	13049	0.08	0.09
	167-D IV	11681	-	-
	50-69	531	1.91	0.20
	D5	12296	-	-
	126-7	9967	0.03	0.12

## Data Availability

All data are available from the corresponding authors upon request. FCS files are available at https://data.mendeley.com/datasets/k3kjtcybfh/1, accessed on 3 June 2024.

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
