# Peer review of "Applying Flow Virometry to Study the HIV Envelope Glycoprotein and Differences Across HIV Model Systems"

_viruses, 2024, doi:10.3390/v16060935_

Round 1

Reviewer 1 Report

Comments and Suggestions for Authors

In this study, Burnie and colleagues used flow virometry to characterize the conformation of the HIV-1 Env protein on the surface of viral particles. They developed a protocol that does not require fluorescent labeling of virions or other technical strategies to detect viral particles in the supernatant of producer cells. They utilized a panel of 85 different anti-Env antibodies, targeting various regions of the viral protein. Additionally, they compared viral particles produced by different cell types and analyzed the effect of soluble CD4 to demonstrate their ability to detect, by flow virometry, different conformations of Env on the surface of virions.

The manuscript is well written, and the data are convincing and clearly presented.

I only have a few minor comments that could improve this work:

- More details in the materials and methods would be helpful for reproducing the results presented. Given the high background associated with flow virometry, did the authors use filtered media and reagents for their viral productions and staining? What type of filter did they use?

- Is it possible that the viral particles that stain negative with one or more antibodies correspond to defective viruses, naturally present in every viral preparation? Is there a correlation between the infectivity of the viral preparation and the quality of the staining?

- It would be important to test if their protocol is also effective in staining the Env proteins displayed on the surface of transmitted/founder viruses, as it has been shown that they differ from laboratory-adapted strains, for example, in glycosylation levels or the length of the variable loops.

- It is not clear if this protocol also detects extracellular vesicles produced by infected/transfected cells having a size similar to that of the viral particles and thus falling in the gate of analysis. How much the results could be affected by this background ? A comment on this should be added.

- Lines 306-309: the explanation of why they were unable to detect staining on PV is not clear. It would be useful to test PV produced using other Envs. Additionally, is the infectivity of the PV comparable to that observed with an IMC dyspalying the same Env and that can be detected by flow virometry ?

- Fig. S3 D: it is unclear how it is possible to achieve 100% infection using a delta-Env virus.

- Figure 6: display results as a histogram to better visualize the improvement of the staining that is mentioned in the text but is not really visible in the dot plots presented.

Reviewer 2 Report

Comments and Suggestions for Authors

The study meticulously optimizes the flow virometry (FV) technique to detect HIV-1 using various bNAbs and other mAbs. The authors demonstrate that their technique can 1) successfully detect virus env in crude cell supernatant 2) identify conformational changes in the Env protein and 3) estimate the number of Env proteins on the virus when the bNAbs are directly conjugate to PE. A significant drawback of FV is the inadvertent detection of extracellular vesicles (EVs); however, this can also provide insights into EV-related questions in HIV-1 infection. Notably, the authors observe differences in sensitivity to bNAbs across viruses produced from various cellular models of virus infection. This technique raises several intriguing questions regarding HIV-1 vaccine development, such as the use of specific cell lines for virus production. A major concern is that the sensitivity of an antibody to detect different viruses in FV could not be validated using other classical assays like virus-capture and neutralization assays. Therefore, it is crucial to determine if the observed differences are technique-specific artifacts. Overall, the study is novel and will stimulate much discussion in the field.

Comments:

1) Did the authors test the infectivity of viruses produced from different T cell lines (IIIB from Jurkat, H9, and PM1), and can it correspond to the differences in staining with different bNAbs?

2) Figure 4: Can the authors compare the neutralization activity of PGT126 between the IIIB viruses produced from different T cell lines (H9, PM1, Jurkat)?

3) Can the authors comment on whether the gated area might include extracellular vesicles, and if so, to what extent? Additionally, can the authors discuss whether the differences in staining patterns observed in viruses produced from different cell lines could result from variations in the amount of EV production from these cell lines?

4) Testing env mutations leading to conformational changes in the Env trimer is necessary to provide strong support for the statement the FV can detect conformational changes in HIV-1 Env glycoprotein

Round 2

Reviewer 1 Report

Comments and Suggestions for Authors

The authors convincingly answered my previous concern. However, a comment must be added to the manuscript regarding the T/F. Due to the differences between these and the laboratory strains at the Env level, detection by flow virometry could be more challenging.

Author Response

We appreciate the Reviewer's attention to this clarity.

We have added this new text to the Discussion (lines 418-420):

Furthermore, performing flow virometry-based Env characterization on a broader range of diverse viral isolates, including transmitter-founder strains, is an important future direction to determine how differences in Env sequences and post-translational modifications are detected with our unique assays used herein.

Reviewer 2 Report

Comments and Suggestions for Authors

The authors have addressed all my concerns. 

Author Response

Thank you!